# One Hundred Years of Coat Colour Influences on Genetic Diversity in the Process of Development of a Composite Horse Breed

**DOI:** 10.3390/vetsci9020068

**Published:** 2022-02-06

**Authors:** Carmen Marín Navas, Juan Vicente Delgado Bermejo, Amy Katherine McLean, José Manuel León Jurado, Antonio Rodríguez de la Borbolla y Ruiberriz de Torres, Francisco Javier Navas González

**Affiliations:** 1Department of Genetics, Faculty of Veterinary Sciences, University of Córdoba, 14071 Córdoba, Spain; v32manac@uco.es (C.M.N.); id1debej@uco.es (J.V.D.B.); 2Department of Animal Science, University of California Davis, Davis, CA 95617, USA; acmclean@ucdavis.edu; 3Centro Agropecuario Provincial de Córdoba, Diputación Provincial de Córdoba, 14071 Córdoba, Spain; jmlj01@dipucordoba.es; 4Unión Española de Ganaderos de Pura Raza Hispano-Árabe, 41001 Sevilla, Spain; uegha@caballohispanoarabe.com

**Keywords:** composite breed, Spanish purebred horse, Arabian purebred horse, coat colour, genetic diversity

## Abstract

Genetic diversity and demographic parameters were computed to evaluate the historic effects of coat colour segregation in the process of configuration of the Hispano-Arabian horse (Há). Pedigree records from 207,100 individuals born between 1884 and 2019 were used. Although coat colour is not a determinant for the admission of Hispano-Arabian individuals as apt for breeding, it may provide a representative visual insight into the gene contribution of Spanish Purebred horses (PRE), given many of the dilution genes described in Há are not present in the Arabian Purebred breed (PRá). The lack of consideration of coat colour inheritance patterns by the entities in charge of individual registration and the dodging behaviour of breeders towards the historic banning policies, may have acted as a buffer for diversity loss (lower than 8%). Inbreeding levels ranged from 1.81% in smokey cream horses to 8.80 for white horses. Contextually, crossbred breeding may increase the likelihood for double dilute combinations to occur as denoted by the increased number of Há horses displaying Pearl coats (53 Há against 3 PRE and 0 PRá). Bans against certain coat colours and patterns may have prevented an appropriate registration of genealogical information from the 4th generation onwards for decades. This may have brought about the elongation of generation intervals. Breeder tastes may have returned to the formerly officially-recognised coat colours (Grey and Bay) and Chestnut/Sorrel. However, coat colour conditioning effects must be evaluated timely for relatively short specific periods, as these may describe cyclic patterns already described in owners’ and breeders’ tastes over the centuries.

## 1. Introduction

Coat colour may have played a major role during the early domestication events and initial selection of the domestic horse [1]. Colour and marking patterns would not only become a relevant practical tool for individual identification [2,3], but also, for the confirmation of allelic segregation between parents in the era of genetics. This enables the distinction among the individuals of particular breeds [4]. The diversity of the coat colours of Spanish ancient horses was depicted in ancient paintings and referenced in literature [5]. Roans and diluted coats, and certain white markings such as appaloosa, piebald or skewbald, have been described in the art and literature from many centuries ago.

The improved features derived from heterosis resulting from the cross between PRE and PRá horses were already acknowledged in the 12th century by the author Ibn al-’Awwam. In the first section of the thirty second chapter of its Kitāb Al-filāḥaẗ (Book of agriculture) [6], the author not only describes coat nomenclature, but also grades the repercussion of coat particularities on the functionality and behaviour patterns of horses (the benefits of bay and certain light coated horses (white or grey ones), among others).

Additionally, problems linked to after-birth markings and the detrimental conditions inherent to mixed colour patterns were addressed. Spanish testimonies from the 15th century reinforce such a hypothesis. For instance, *De la naturaleza del cavallo* by Fernández de Andrada [7], attributed preferable aptitudes and performance (docility, resilience, bravery and velocity) to silver and very dark horses, always considering markings or colour mixtures desirable [5].

Despite a posterior unlinking [2], the conditioning effects of coat colour on the performance of Arabian horses were already described in 1524 AD in the North African manuscript Kitab Al-Ihtifal Fi Istifa’ Ma Lil-Khayl (Treaty of celebration of the achievements of the horse) by Abu Abdullah Muhammad bin Muhammad bin Ahmad bin Juzayy al-Kalbi (1357 AD) [5].

This compilation of hadiths, sayings and poetic verses describes the qualities that should be expected from horses presenting certain coat colour phenotypes and shares its main topic with other unpublished manuscripts dating back to the 16th century such as *Razze del Regno* by Federico Grisone [8,9], which serves as a witness of the interbreed relevance provided to phaneroptical features. In this regard, Stachurska and Ussing [10] state that the research to date on the racing performance of horses of different colours has considered solely two loci controlling the colour, hence, future deeper knowledge of the molecular genetic background of performance may give an unequivocal answer whether such relationship exists and, if so, which particular traits are associated with which colours. For instance, a recent work by Junqueira, et al. [11] reports that coat color influence might be explained as a pleiotropic effect of the genes that cause this phenotypic variation, influence morphometric measures and, by extension, performance.

PRE national and international market fluctuations promoted the decline of Há breed until it became a minority rare breed by the mid-1980s [10]. Even though the widely acknowledged enhanced characteristics of Há led to the event of constitution of its studbook in 1986 and its later official recognition as an special protection breed (Official Catalogue of Spanish Breeds, APA Order/2129/2008) [11], the pivotal role it played in military campaigns during the mid-19th century led to the breed standardization in 2002.

Coat colour genetic control may be compulsorily considered in the breeding programmes of horse breeds with fixed standards [12]. Contextually, although no explicit mention of coat colour is made in the Há breed standard [13], multiple conditioning factors (breeder preferences among others), may have shaped its population structure and genetic diversification process, and also those occurring in its two ancestral breeds [14,15].

Even if crossbreeding offered the opportunity to combine the desirable characteristics of Há’s two ancestral breeds, PRE and PRá, potentially increased performance due to the complementary effects of heterosis [16]. Still, the planning of matings should be a priority. In these regards, Cervantes, et al. [17] emphasized including individuals’ complete genealogies to attain reliable genealogical parameters (such as inbreeding). This may be even more important in composite crossbreeds, in which ancestral breeds participating of the cross do not equally contribute to it (heterosis imbalance), as reported in Há horses [10].

Pedigree information-based methods have been reported as efficient tools to evaluate diversity, especially when the use of other tools such as molecular markers (SNPs or microsatellites) is not feasible due, for example, to economic reasons. Improving pedigree robustness (close to 100% completeness) buffers the potential inaccuracy of pedigree-inferred genetic parameters when computed after incomplete genealogical information [18]. Furthermore, even if large sets of genomic markers (microsatellites or SNPs) and empirical estimates of relatedness have been reported to improve pedigree-based approaches, the lack of knowledge of the historic allele frequencies present in the population hinders the efficiency of molecular techniques, as these may not distinguish between the identity by descent (IBD) and identity by state (IBS) probabilities that support genetically mediated similarities among relatives. Indeed, this produces a bias in diversity estimations (as a consequence of genetic drift or unknown bottlenecks occurring throughout the history of a population). For these reasons, methods should be tailored and preferably combined, for the specific context of each population to maximize the reliability of the estimation of diversity parameters.

For crossbreeding systems to be effective, any potential conditioning factors must account for such unbalanced contribution to hybrid vigour [16]. In this context, market discrimination against certain breeds or colours (the lack of uniformity in colours or the distribution of white markings), or in favour of those linked to an added value, has often been reported to condition breeders’ aptitudes, breeding tendencies or even husbandry patterns [19,20,21].

Coat colour knowledge of the mechanisms of inheritance is useful for coat pattern selection, but also for the induction/introduction of new patterns to already established breeds. Such features may be linked to desirable functional traits or their perception by breeders/judges [22,23,24], enabling a parallel upgrade with commercial trends or to undesirable diseases [4,19], permitting early prevention of detrimental conditions [16,21].

Whereas a major (single) gene may regulate the expression of coat colour, and single genes may modify such expression, the process is generally quantitative [25], complex and often breed-specific. The ability to maintain complete control of coat colour conversely reduces decreases as the number and diversity of breeds involved in crosses increase. In this context, apparently equal colour phenotypes may depend on a completely different genetic background [26,27].

For this, determining parental coat colour genetic code must be a prior consideration to selection of breeding pairs. From a genetic standpoint, patterns expression regulated via homozygosity of recessive alleles may evidence the particular contributions of breeds and may be considered ‘breed true’ signs as they reveal the underlying genetic mechanisms involved in crossbreeding [28]. In this context, although it is true that most horse breeds share most alleles, few horse breeds have little bits of this information that are absolutely unique to them; hence, some combinations can more likely pop up and be typical of certain breeds than of others [29]. This leads to the conclusion that the more is that is known about coat colour inheritance patterns of the parental breeds participating in a composite breed, the more effective mating plans seeking a particular outcome, either functional or aesthetical, will be.

To this aim, the main objective of this study was the evaluation of the effects of coat colour on genetic diversity and population structure in the Hispano-Arabian horse and its two ancestor breeds. Genetic and demographic parameters were calculated to measure the trends described by the existing gene flow and quantifying the risk of genetic diversity loss throughout the history of development of the three breeds.

## 2. Materials and Methods

### 2.1. Pedigree Registries and Software for Genetic Analyses

The historic pedigree records of Há, PRE and PRá breeds were considered in this study comprising a database consisting of 207,100 horses. A full description of the sexes and coats distribution in the historic and current datasets is presented in Table 1 and Figure 1 and Figure 2. Appendix A presents a summary chart of the genetic background of coat colour phenotypes described in PRE, PRá and Há horse breeds to date. Genetic analysis was conducted on the historical dataset comprising all animals (dead and living) and on a filtered dataset containing only living animals to evaluate the evolution and trends described by diversity and population structure parameters. ENDOG (v4.8) software [29] was used for all the analyses except for the analysis of ancestral contributions and probabilities of gene origin which were performed using CFC version 1.0 [30].

### 2.2. Genealogical Information

The pedigree completeness index (PCI) summarizes the proportion of known ancestors at each ascending generation through the analysis of generations. The analysis of generations comprises the evaluation of maximum number of traced generations, the number of complete traced generations, the number of complete equivalent generations, calculated as (1/2*n*) where *n* is the number of generations setting the individual apart from each known ancestor, equal to $\overset{n_{b}}{\underset{a=1}{\sum }}\frac{1}{2^{g_{ab}}}$, where *n_b_* is the total number of ancestors of the animal, *b* and *g_ab’_* is the number of generations between *b* and its ancestor *a* [31], and the pedigree information quality assessing the proportion of pedigree registered known parents, grandparents, great-grandparents and great-great-grandparents.

Generation intervals [32], and the mean age of parents when their offspring were born were calculated for the four gametic pathways (stallion to colt and filly and mare to colt and filly) across coat colour possibilities. The stallion/mare ratio was calculated considering the percentage of mares and stallions with breeding progeny and the number of breeding animals selected.

### 2.3. Inbreeding, Coancestry, and Nonrandom Mating Degree

Individual inbreeding (F) was computed using the methods in Meuwissen and Luo [33]. Each individual’s average relatedness (AR) was calculated as Gutiérrez, et al. [29]. According to Leroy, et al. [34], F and coancestry (C) coefficients are identity estimators by descent (IBD), a probability that differs whether the alleles considered belong to a single individual or two individuals, respectively. The individual rate of inbreeding ($\bar{\Delta F}$) for the generation was calculated according to Gutiérrez, et al. [35] by $\Delta F_{b}=1-\sqrt[{t_{b}-1}]{1-F_{b}}$, where t_b_ is the number of complete equivalent generations and F_b_ is the inbreeding coefficient of the individual *b*. The individual rate of coancestry ($\bar{\Delta C}$) for the generation was computed following the methods described by Cervantes et al. [36] by $C_{ba}=1-\sqrt[{\frac{tb+ta}{2}}]{1-C_{ba}}$, where t_b_ and t_a_ are the number of equivalent complete generations and C_ba_ is the coancestry coefficient for the individuals b and a. The degree of assortative mating (non- mating of individuals having more genetic or phenotypic traits in common than likely at random or disassortative mating), was computed following the methods in Caballero and Toro [37], by (1−F)=(1−C)(1−α) [38].

### 2.4. Ancestral Contributions and Probabilities of Gene Origin

The effective number of founders (*f_e_*), was calculated using $f_{e}=\frac{1}{\sum_{k=1}^{f}q_{k}^{2}}$, where *q_k_* is the probability of gene origin of the *k*th founder and *f* is the real number of founders [39]. The effective number of ancestors (*f_a_*), was determined by $f_{a}=\frac{1}{\sum_{k=1}^{f}p_{k}^{2}}$ where *p_k_* is the marginal contribution of a *k*th ancestor [31]. The effective number of founder genomes (*f_g_*) was computed as the inverse of twice the average coancestry as reported in Caballero and Toro [37]. The expected marginal contribution of each major ancestor *j* was computed as by Boichard, et al. [31], and the contributions to inbreeding of nodal common ancestors (inbreeding loops), were computed according to Colleau and Sargolzaei [40]. The mean effective population size ($\bar{N_{e}}$) [38] was calculated as $\bar{N_{e}}=\frac{1}{2\bar{\Delta IBD}}$. The number of equivalent subpopulations was computed according to Cervantes, et al. [41] using $S=\frac{\bar{N_{e}C_{i}}}{N_{e}F_{i}}$, where $\bar{N_{e}C_{i}}=\frac{1}{\left(2\bar{\Delta C}\right)}$, is the mean effective population size considering the coancestry coefficient [42] and $\bar{N_{e}F_{i}}=\frac{1}{\left(2\bar{\Delta F}\right)}$, considering the inbreeding coefficient. The rate of loss heterozygosity due to inbreeding per generation measured by F is equal to 1/(2*N_e_F_i_*), where N_e_ is the effective population size. Genetic diversity (*GD*), was calculated using $GD=1-\frac{1}{2f_{g}}$. GD lost in the population since the founder generation was estimated using 1−GD. GD loss derived from the unequal contribution of founders was estimated as Caballero and Toro [37] using $1-GD^{*}$, where $GD^{*}=1-\frac{1}{2f_{e}}$. The difference between *GD* and *GD** indicates the *GD* loss owed to genetic drift accumulated since the foundation of the population [39], and the effective number of non-founders (*N_ef_*) was computed using $N_{ef}=\frac{1}{\frac{1}{f_{ge}}-\frac{1}{f_{e}}}$ considering the formula proposed by Caballero and Toro [37]. CFC version 1.0. was used to perform the analysis of ancestral contributions and probabilities of gene origin [30].

Distance-based tree construction algorithms are supported on the calculation of genetic distances, as in [43] F_ST_ or Nei’s [44]. In particular, Nei’s genetic distance were used to construct the trees in this study given they represent “raw distances” among subgroups. The use of F_ST_ or Nei’s may lead to a similar interpretation. In these regards, when comparing breeds of the same species, F_ST_ values are always expected to be below 0.05, as this may be the lower limit for species differentiation. However, when breeds are connected (PRE and PRá are the parental breeds of Há) the values slightly increase over 0.05 [45]. Nei’s minimum genetic distances [44] among all the coat colour subgroups were computed. Dendrograms were constructed using the construct Unweighted Pair-Group Method using Arithmetic averages (UPGMA) Tree task from the Phylogeny procedure of MEGA X 10.0.5.

## 3. Results

### 3.1. Pedigree Evolution

The number of individuals born progressively increased in PRá, PRE, and Há horses from 1944 to 2006 when a sharp decrease occurred (Figure 1). The proportion of yearly born animals across coat colour subgroups and sexes maintained throughout 119 years of history, with grey and bay horses representing the most numerous fractions across the three breeds analysed, and 88.03% of the whole population considered in the present study (Figure 2 and Figure 3). Even if proportions were maintained, number of births was historically considerably higher in PRE horses when compared to PRá and Há horses [10,46,47]. When the distribution of coat colours across breeds was evaluated, PRá and Há proportions had similar patterns, with a slightly higher representativity of chestnut/sorrel individuals in PRá horses. Contrastingly, bay and grey animals were remarkably more numerous than those displaying one of the remaining coat colours in PRE horses, with proportions of black animals being slightly higher than those reported for PRá and Há horses.

The number of complete generations ranged from 1.62 to 5.01, with the lowest values in the rank being reported by cremello and diluted coats such as pearl and isabelline. Contrarily, higher values in the range were reported for grey and saturated colours such as black, bay, white or chestnut/sorrel. Values slightly increased in the current population, except for cremello, pearl and isabelline coats. The same pattern was described for the number of equivalent generations with these being two generations higher in the aforementioned diluted coats and three in the rest. Average numbers of equivalent generations [48] converged in the historic and currently living populations across saturated coat colour subgroups while average equivalent generation numbers decreased in diluted coat colours subgroups (Table 2).

The presence of incomplete pedigrees is characteristic of older animals for which the control of genealogy may not have been carried out as strictly as it occurs in recent individuals. The increase in pedigree completeness, and indirectly in complete and equivalent generations from the historic to the current population, may derive from the fact that animals with incomplete pedigrees may have disappeared; hence, they are no longer considered to compute the values of diversity parameters in the currently living population.

The number of complete generations [48] was around half the number of equivalent generations, which suggests that even if the genealogical information of individuals has progressively increased throughout the years, incomplete and partially incomplete pedigrees are still representative in the population. However, the presence of partially incomplete pedigrees may be unequally distributed across coat colour subgroups. For instance, as reported in Figure 4, coat colour subgroups such as grey, bay, black, overo and roan maintain pedigree completeness levels of over 80% at the fifth generation. Chestnut/sorrel subgroup have pedigree completeness levels of around 70% at the fifth generation. While cremello, isabelline and white coat colour subgroups presented pedigree completeness levels ranging from around 40 to 60% at the fifth generation, the lowest pedigree completeness levels were reported by palomino and pearl coat colour subgroups, with completeness levels of slightly over 35% at the fifth generation.

Generation intervals and mean age of parents when their offspring were born were similar for each coat colour subgroup with values around 10 years (Figure 5).

Cremello and isabelline coats, respectively, doubled and tripled the aforementioned values. A summary of the results derived from offspring analyses (Appendix A) reports the number of foals displaying a grey coat colour to be considerably higher than those displaying any of the rest of coat colours. However, the historic numbers of grey foals dramatically decreased eight times in the currently living population. Bay, chestnut/sorrel and black foals were the most numerous across all coat colour possibilities, with their historic numbers currently maintained. Average number of foals per mare was almost constant and around two foals per mare across coat colour subgroups except for cremello, pearl and roan coats, for which values were half the aforementioned. These trends were not reported in average number of offspring per stallion, for which rather different situations were reported.

The highest average numbers of offspring per stallion (over 11 foals/stallion) currently maintaining historic numbers were reported in bay, black, dun and cremello coats. Additionally, grey and isabelline coat colour subgroups showed an increase in average number of foals per stallion. The opposite trend was reported by this parameter in chestnut/sorrel and Roan with values slightly decreasing in current population. This decrease was drastic in average foal number per stallion in the Overo coat colour subgroup (Appendix A).

Percentages of offspring selected for breeding from mares were on average 10% higher than those for offspring selected for breeding from stallions. As suggested by Appendix A, all percentages for offspring from stallion selected for breeding have slightly (grey, chestnut/sorrel, black, isabelline, cremello and roan) or moderately (white and pearl) decreased or maintained (dun) in the current population, except for the offspring selected for breeding from stallions presenting a bay coat colour. All percentages of offspring from mare selected for breeding slightly decreased in the current population of all coat colour subgroups, except for palomino coated mares for whom percentages were historically maintained.

### 3.2. Inbreeding, Coancestry/Kinship and Degree of Non-Random Mating

Table 3 presents the number of inbred and highly inbred animals. Animals presenting any level of inbreeding different from 0 were considered to be inbred. However, as suggested by Beuchat [49], even if the deleterious effects of inbreeding begin to become evident at an inbreeding level of around 5%, it is when inbreeding reaches 10% that there is significant loss of vitality in the offspring as well as an increase in the expression of deleterious recessive mutations. Hence, animals presenting values over 10% for inbreeding were considered to be highly inbred animals.

Historic inbreeding levels for coat colours subgroups are represented in Figure 6 and Figure 7 and ranged from 6.25 to 8.33%, except for palomino, pearl and smokey cream coat colour subgroups, for which inbreeding levels were 4.75, 2.86 and 1.81%, respectively.

Current inbreeding values range from 6.95 to 8.80%, with palomino, pearl and smokey cream coat colour subgroups maintaining at 4.75, 2.91 and 1.81%, respectively. The individual increase of mean inbreeding (ΔF), was at or slightly surpassed the critical limits of 1%, except for pearl, palomino and smokey cream, which did not reach 1% and cremello and isabelline coat colours subgroups that almost doubled the 1% level (Figure 6). All coat colour subgroups reached levels of inbreeding over 10% (Figure 7) for Arabian and Spanish Purebred horses throughout 120 years of history, which occurred at the beginning of registries (1980s) in the Há horse breed for black, bay, dun, isabella, roan and grey coat colour subgroups, and which also occurred again from 2019 onwards.

Maximum inbreeding coefficient values ranged from around 45 to 50% in grey, bay, chestnut/sorrel and black coat colour subgroups. The other coat colour subgroups had values of maximum inbreeding coefficient ranging from slightly above 11 to around 37%. The lowest maximum inbreeding coefficient (1.81%) was reported by smokey cream animals.

The historic percentage of inbred animals increased in the current population of each coat colour subgroup. Although the percentage of inbred animals widely varied across coat colour subgroups from 1% to 98.01% of the horses in a particular coat colour subgroup being inbred, a very broad range for average coancestry between 1.16 and 5.23% was found (Table 3).

Nonrandom mating degree ranged between −0.02 to 0.06, historically maintaining the same levels except for the white coat colour subgroup, for which it increased from 0.01 to 0.04, Chestnut/sorrel, which slightly increased from 0.05 to 0.06, and overo, which slightly decreased from 0.05 to 0.04 (Table 3). The lowest values (−0.02) were reported for smokey cream coat colour subgroup (Figure 6). GCI was around 4 to 6 in diluted coat colour subgroups (isabelline, pearl and cremello), while in saturated coat colours subgroups GCI values ranged from 8 to 11 (Figure 6). A particularly high value was shown for the smokey cream subgroup with a GCI of 15.07 (Table 3).

### 3.3. Ancestral Contributions and Probabilities of Gene Origin

The representativity of founders and ancestors has drastically decreased in the current population, with saturated coat colours presenting the highest historic numbers of founders, but also the greatest proportional decrease in them. Grey, bay and chestnut/sorrel coat colours subgroup presented the highest number of founders and ancestors. However, these subgroups had a five times smaller number of founders and ancestors in the current than in the historic population. This five-times lower number of founders or ancestors was also present in white and roan coat colour subgroups. A slighter decrease in the number of founders or ancestors was reported for Black, Dun and Overo coat colour subgroups, while diluted coats such as isabelline, cremello and pearl historically maintained the number of founders and ancestors up until the present. No founder was historically found in the palomino or smokey cream coat colour subgroups, and only 19 ancestors and one ancestor were, respectively, found for these coat colour subgroups (Table 4).

The ratio between effective number of ancestors (*f_a_*) and the number of founder equivalents (*f_e_*) in the historic population set ranged between 0.46 and 0.68, and decreased to 0.39 and 0.66 in the current population, with values for almost all coat colour subgroups being around 0.5 and 0.6. The ratio between the number of founder genome equivalents (*f_g_*) and founder equivalents (*f_e_*) ranged between 0.23 and 0.39 in the historic population, and 0.19 and 0.36 in the current population, with values around 0.2 and 0.4 in almost all coat colour subgroups.

Numbers of founder genome equivalents (*f_g_*) translated into higher than 92–93% levels of genetic diversity (lower than 7–8% levels of genetic diversity loss) as reported in Table 5, for historic and currently living populations for each of the coat colour subgroups. As an exception, the smokey cream subgroup accounted for 49% and 51% of genetic diversity loss in historic and current populations, respectively.

The highest genetic diversity levels were found for the chestnut/sorrel and overo coat colour subgroups (96–97%), for which the number of ancestors explaining 50% and 75% of the gene pool doubled those in the rest in coat colour subgroups. Genetic drift since founders was responsible for from 1% to 2% of genetic diversity loss in the historic and current populations of all coat colour subgroups, with the exception of the white coat colour subgroup, for which it historically reached 4%.

These percentages increased to 3% to 6% with the exceptional levels of 50% found for smokey cream coat colour subgroup when genetic diversity loss was computed considering the effects derived from bottlenecks, genetic drift and the unequal contribution of founders. This may be supported by only one ancestor explaining the total of the diversity in this coat colour subgroup. Twice the number of ancestors to explain 75% of genetic diversity in saturated coat colour subgroups was needed when compared to the number of ancestors needed to explain 75% of genetic diversity in diluted coat colour subgroups.

The number of equivalent subpopulations reported in the currently living population was always below or equal to 0.4. The effective size of the population calculated through the individual inbreeding rate was from three to fourteen times higher than when it was calculated through coancestry rate (Table 6). The effective population size calculated through the individual inbreeding rate in the historic and current population sets was always close to 50, doubled this critical value in the case of the grey coat colour subgroup, and almost doubled it for the palomino coat colour subgroup. Cremello and pearl coat colour subgroups were the only ones for which *N_e_F_i_* did not reach the critical limit of 50, which may save them from immediate risk of disappearance. By contrast, the effective population sizes calculated through the individual coancestry rate were always lower than 22 individuals.

### 3.4. Genetic Relationships between Coat Colour Subgroup

The average Nei genetic distances among coat colour subgroups in historic and current populations was 0.004 and 0.014, respectively (Figure 8). The mean historic and current coancestry within subpopulations, when the criterion for subdivision was the coat colour, was 0.052 and 0.040, respectively. Mean historic and current coancestry levels in the metapopulation when the coat colour was chosen as the population subdivision criterion were 0.048 and 0.026, respectively.

With regards to Wright’s F statistics (Table 7), the inbreeding coefficient relative to the total population (F_IT_) was 0.033 for the historic population and 0.032 for the currently living population when coat colour was chosen as the subdivision criteria. The inbreeding coefficient relative to the subpopulation (F_IS_) varied from 0.029 for the historic population to 0.018 for the current population (Table 7). The correlation between random gametes drawn from the subpopulation relative to the total population (F_ST_) was 0.004 for the historic population and 0.014 for the currently living population.

## 4. Discussion

Despite the remarkably higher PRá contribution to the gene pool of Há horses (70% PRá, 30% PRE) reported by Marín Navas et al. [10], our results suggest coat colour allelic richness may presumably be ascribed to a PRE origin. Several bottlenecks have occurred throughout the history of PRE population since the PRE studbook was closed and the breed became genetically isolated in the 1880s [50]. After the 1880s, several bottlenecks continued occurring due to the effects of a century of wars that concluded with The Spanish Civil War at the National level (1936–1939) and World War I (1914–1918) and II (1939–1945) at the international level. The drastic reduction in the number of PRE effectives, and the profitable obtention of mules, led to the development of several policies to recover and protect the breed. Among them, low numbers promoted the proclamation of bans against free crossing between jackstocks and Spanish mares under death penalty in Andalusia, Extremadura and Murcia (below the “Royal Line”) from the 13th to 19th centuries [51], and against exportations until the early 1960s. However, despite these attempts, numbers may only have recovered after the first African horse sickness outbreak disappeared in 1966 [52] and after the end of Francoist dictatorship in 1975 [53].

Contrary to PRá, almost all possible coat colours are currently present and are represented in the early volumes of PRE studbook, except for spotted and white patched coats. However, coat colour relative frequencies have changed over time. For instance, although animals presenting grey gene linked progressive greying patterns are currently numerous, they were originally a small minority and their numbers did not increase until the 1970, when the ban against other coats started. Such a reduction in colour phenotypes brought about a 32 years period of loss of diversity (bottleneck), derived from the attempts to eliminate every coat colour except for grey, bay and black from the PRE studbook [54] (Figure 1). This colour banning policies applied to horse breed standards over the years may have affected not only the way horses have been bred, but how they may have been registered with false or incorrect information, or even remain unregistered. However, this problem is not unique to Spanish horse breeds; for instance, the Cleveland Bay Horse studbook in which purebred animals that were born chestnut were historically not registered or registered as part-breds. However, this situation has been addressed at present under current EU Zootechnical Legislation; this practice is forbidden and animals need to be compulsorily registered as purebred in the studbook even if they may be registered as animals whose “coat colour does not conform to breed standard”.

The main aim of the early period of the PRE breed was to uniformly make grey and bay coat colours the standardized patterns. For this, not only matings were directed, but also widely different coat colours, such as buckskins and blacks, were registered as bay. Additionally, the Spanish Studbook of the PRE has always treated grey as a separate coat, ignoring the actual basecoat of the individuals (Appendix A).

These circumstances led to the fact that after the beginning of the World War II (Figure 1), the number of breeding animals (studs and mares) were predominantly grey (with their base coat colour being unknown). In conclusion, 75% of PRE carried at least a single incidence of the G allele from 1880s until 2007, a period during which PRE studbook was controlled by the Spanish Ministry of Defence. Although these events could be thought to mislead the analysis of coat colour distribution across the PRE pedigree, they ensured genetic diversity from chestnuts, duns, buckskins, blacks, and other coats which greyed out very quickly, being preserved in the studbook as bay and grey.

Bottlenecks and breeding policies along the history of PRE horses led to the overrepresentation of phenotypes covering characters of certain coat patterns, such as grey gene encoded patterns. Grey gene homozygous animals can never produce anything except foals that are grey. In this context, recovering the underlying colours is not only challenging because of the intrinsic difficulty in achieving it, but also may imply the loss of quality and functionality. As a consequence, phenotype misidentification makes coat colour determination more complex as the systematic large-scale determination of genetics may be cost-demanding.

The instauration of APA Order/3319/2002 [54] derogated three decades of restrictions from 1 January 2003 onwards, to permit the registration of any coat colour. This event started a yearly, slight progressive increase in the numbers of mainly chestnut/sorrel individuals (Figure 1). However, an earlier certain census recovery could be presumed from 1980s onwards, two decades before prohibition was derogated, due to the under-cover labour and interest of breeders [54]. PRE breeders, aware of the fact that the Spanish Government was going to change the Law, collectively started to preserve forbidden coloured PRE foals, which had been sold as undocumented and ‘nonpedigreed’ to working homes.

Afterwards, public administration offered breeders two years during which any adult horse presenting a previously banned colour could be registered if its parentage could be proved by DNA testing, pedigree, and provenance, which enriched the diversity present in the PRE population. From the beginning of the second decade of the 21st century, the quality of these unrepresented colour coats improved enormously due to the increase in numbers enabling more efficient breeding practices for better type and conformation, and also driven by owner tastes and changes occurring in an always fluctuating market [14], which indirectly enhanced the diversity of derived breeds in whose development PRE participated, such as in Há horses.

The examination of a dendrogram constructed from Nei genetic distances (Figure 8) and Wright’s statistics (Table 7) suggests the existence of a clear population structure across coat colour subgroups even if Há and its two ancestral breeds are considered together, which may still be partially supported by the relationships that have been established between the two ancestor breeds along the process of development of the Há [10]. In this context, proximity between the breeds was evidenced when PRá was compared to PRE horses [10]. This finding derives from the fact that Há hores can have a up to 75% of PRá blood.

As suggested in Alanzor Puente et al. [45], an F_ST_ of one may imply that all genetic variation is explained by the population structure, mainly conditioned by the existence of barriers to gene flow (geographical, linguistical, sociocultural, and even economical) and, therefore, that the three populations examined would not share any genetic diversity. However, our results suggest the contrary, because when comparing breeds of the same species, F_ST_ values are always expected to be below 0.05, as this may be the lower limit for species differentiation.

With this being said, when breeds are connected (PRE and PRá are the parental breeds of Há) the values slightly increase over 0.05. Hence, computing F_ST_ values can report very important information about the relationships among lower-scale genetic subdivisions of a population, such as breeds or varieties, or those linked to specific features such as coat color or even functionality.

This becomes even more patent when values for F_ST_ are comparatively interpreted with F_IS_ values. At a breed level (F_ST_ below 0.05 context), when coat colours are considered as the criterion for population subdivision, F_IS_ negative values may address the existence of a disequilibrium in breeding policies acting in favour of an unexpected mating rate of different coat animals under a model of random mating (Table 3). 

A diluted coat colour subgroup cluster (0.11 distance from origin) has formed between cremello (C^prl^/C^Cr^), isabella/isabelline (C^prl^/C^prl^) and pearl/perlino horses (C^Cr^/C^Cr^ or C/C^Cr^) (Figure 8). This clustering pattern may be linked to the inheritance cream/pearl genes whose expression is associated to the Solute Carrier Family 45 Member 2 or Membrane-associated transporter protein (MATP) gene, which are variants located in the chromosome 21. A distance of 0.007 was reported between cremello (C^prl^/C^Cr^) and Isabella/Isabelline (C^prl^/C^prl^) while pearl/perlino horses (C^Cr^/C^Cr^ or C/C^Cr^) are 0.013 and 0.018 apart from the aforementioned coat colour subgroups, respectively.

The dominant cream gene has been reported to activate the pearl phenotype if the two variants are present (C^prl^/C^Cr^) [55]. The interaction between compound heterozygous for the pearl C^prl^ and cream C^Cr^ alleles (Appendix A) makes *MATP/SLC45A2* the most plausible candidate gene for the pearl phenotype in horses, as suggested by Sevane, et al. [56]. These authors suggested a missense variation in exon 4 [*SLC45A2*:c.985G>A; *SLC45A2*:p.(Ala329Thr)] as the causative mutation for the pearl coat colour. Additionally, it may likely be involved in the regulation of the expression of cremello, perlino and smoky cream like phenotypes associated with the compound C^Cr^ and C^prl^ heterozygous genotypes (known as cream pearl in the PRE breed). Some PRá horses may appear to be palomino, but are genetically chestnuts with flaxen manes and tails, as the *MATP/SLC45A2* gene is completely absent from the Arabian horse gene pool [57]. For instance, no Cremello (C^prl^/C^Cr^), Isabella/Isabelline (C^prl^/C^prl^) or pearl/perlino horses (C^Cr^/C^Cr^ or C/C^Cr^) was reported in the PRá population considered in this study (Figure 3). This evidence suggests that C^Cr^C^prl^ heterozygous genotype may be present in Há, and may possibly be better called cream pearl than cremello, following the nomenclature in PRE horses, and may fully derive from and Spanish ancestry, to which the origins of the presence of pearl gene diluted coats have been ascribed, as it has been reported for other Iberian breeds such as Losino [58]. This could also be applicable to isabella coat for which a homozygous double dose of C^prl^ is present.

The earliest separation of the smokey cream coat colour subgroup may occur at a 0.26–0.27 distance from the diluted cluster comprising cremello, isabella/isabelline and pearl/perlino, which may be ascribed to the presence of the double dilute dose (C^Cr^/C^Cr^) of *(MATP/SLC45A2)* also present in pearl/perlino horses, but which may require the additional presence of a double homozygous dose of agouti-signaling protein *(ASIP)* and at least one dominant extension allele (E_) of the melanocortin 1 receptor *(MC1R)* gene.

The palomino coat colour subgroup separates from the aforementioned cream/pearl diluted coat cluster at a 0.034 distance. This separation may stem from the presence of a double recessive doses of the Extension gene (ee) in palomino horses, together with the absence of a C^prl^, which can be present in palomino pearl individuals and when occurring may have been classified as a Pearl animal. Palominos and smokey blacks are still quite unusual, probably because the chestnut and black basecoats are in the minority compared to bay coated PREs, but also because breeding practices may have focused on breeding seeking the production of individuals with other coat colours.

Double cream gene dilutes and double pearl gene dilutes are still very rare, as shown in Figure 3. However, composite breeding may increase the likelihood of double dilute combinations as denoted by the increased number of diluted coats, such as pearl. This may be supported by the fact that generation intervals were similar and slightly above 10 years across coat colour patterns, while considerably higher generation intervals were reported for cream pearl (cremello in the studbook) (around 20 years) and isabelline/isabella horses (around 25 years). This colour pattern appears in horses as a consequence of the interaction between recessive alleles; hence, data may have been subjected to unintentional or intentional misrecording throughout history (Figure 5).

The overo coat was valued by Hispanic Muslims, as denoted by ancient texts dating back to the Fitna of al-Andalus (1009–1031), a period of instability and civil war that preceded the ultimate collapse of the Caliphate of Córdoba [59]. Figure 8 suggests overo patterns may preferably appear when the underlying coat colour is chestnut/sorrel (0.010), although the distance between overo coat subgroup and other solid saturated colours such as roan (0.012), grey (0.013), bay (0.015), dun and black (0.018) or white (0.023) suggests overo patterns may appear on any underlying colour of the aforementioned in Há and its two ancestral breeds. For instance, the Spanish word overo (hobero) was used as a base name on almost every horse having more than one color [60]. However, for PRE, the allusions to the overo coat described in the literature, reduces to chestnut/sorrel animals over which overo white markings are present, which has also been described for other Spanish horses, such as Pottoka [61].

This suggests that the potential appearance of other overo/solid colour combinations may be linked to the contribution of PRá horses. Contextually, among the contributions of the Iberian culture, Arabian people developed equestrian nomenclature and would, for instance, call spotted horses, Hejar-el-wad, which means stones of the river. Indeed, the Arabic term hoberi evolved into the Spanish term hovero algo which was used to address spotted horses by Spaniards of the time [60]. This provides evidence of the popularity and transcendence across centuries of the overo coat for Arabian people.

The saturated coat colour cluster may be divided in three ramifications, with the latter of the three subdividing in two branches (Figure 8). The white coat colour was the earliest to separate. The distance of separation (0.007 to 0.012) may stem from the fact that white appears independently of the underlying coat colour due to the action of heterozygous dominant white gene alleles (Ww) and ‘masks’ them. This earlier separation may also be supported on the three times increase of nonrandom mating degree in the White coat colour subgroup, which denotes the increasing affinity of breeder tastes for the coat.

A second cluster forms when the dun coat colour separates from the majority saturate coat colour cluster comprising grey, bay, roan and black at a distance of 0.006 each. The dun coat color involves the regulatory action of the dun gene with dd, d/nd1, or dd being responsible for dun patterns; that is, the presence of primitive markings on any bay, black, or chestnut/sorrel-based horse affected by the dilution gene brightens both red and black pigments in coat colour, lightening the base body coat and suppressing the underlying base colour to the mane, tail, and legs. Primitive markings increasingly fade when ND1/ND1 and ND1/d are present, but do not disappear unless the dd genotype is present.

A third cluster ramifies into two branches which gather together grey and roan and black and bay with a distance of 0.001 between each pair. The black and bay distance is determined by the presence of the recessive genotype of the Agouti gene (aa) which condition the appearance of a solid black coat colour, while the distance between roan and grey, and of the latter with bay and black, may stem from the fact that roan and grey patterns may appear due to the effect of homozygous dominant and heterozygous grey and roan alleles (GG or Gg and RnRn or Rnrn, respectively) on any underlying coat colour. For instance, in the case roan horse’s skin is damaged by even a very minor scrape, cut or brand, its coat will grow back in solid-colored without any white hairs. These regions of solid-colored coat are called “corn spots” or “corn marks” and may appear even without the horse having had a visible injury.

A clear linkage has been reported with chestnut/sorrel, bay and black at the extension locus in horses [62]. If a horse possesses one chromosome with the wildtype non-chestnut allele and the dominant roan allele (*E* and *Rn*), while the other chromosome contains the recessive chestnut allele and the recessive non-roan allele (*e* and *rn*), it will outwardly appear blue roan, barring the influence of other genes. Normally, the chestnut and roan alleles would be separated during chromosomal crossover, but these two linked genes will usually remain together. Such a horse will produce sex cells that are either *E/Rn* or *e/rn*. Mated to chestnut non-roan partners (*e/rn*), the horse would produce primarily blue roans, or chestnut non-roans, but few chestnut roans and few black non-roans. If, on the other hand, the recessive *e* and dominant *Rn* were on the same chromosome, the horse would be expected to produce primarily chestnut roans and non-chestnut non-roans with chestnut, non-roan partners. This linkage is evidenced in Figure 8 by the shortest distances of 0.012 and 0.024 reported between overo and chestnut/sorrel coats and Roan coat colour, and of 0.003 with bay coat, and of 0.004 with Black coat, respectively, when compared to other coat colours.

Furthermore, the roan gene is not present in PRá, even if it officially appears in registries. Roan in PRá is probably associated with sabino. This is evidenced as, unlike with the traditional roan gene, in PRá a roan offspring can be produced by non-roan parents [63]. As a result, a roan coat pattern in the Há horse breed, may derive from the PRE contribution, similar to what has been described for the cream/pearl gene.

Contextually, as suggested by Klungland [64], MC1-R (melanocyte stimulating hormone receptor) allele frequencies vary greatly across breeds, which translates into the expression of different coat colours. In this regard, the same authors report some coats may not even be present in certain breeds, as has been suggested in the literature for breeds evaluated in the present study. This was also supported by Reissmann [65] and Penedo [66], who reported alleles leading to dilutions or patterns that are rare in domestic breeds and not found in some domestic horse breeds and other equine species. Presumably, it is the complexity of genetics of coat colours and the epistatic relationship and existence (or the lack of it) of variants and their frequencies, which result in some phenotypes not being present in the history of the three breeds.

PCI values were very high for the first five generations (from parents to great-great grandparents), which is common to autochthonous horse breeds as supported by Marín Navas, et al. [10], enabling an accurate calculation of genetic diversity parameter. Relatively lower PCI values (lower than 60%) from the 4th generation on were reported for the Pearl, Cremello, Isabelline/Isabella diluted coat colour cluster (Figure 4), which suggests poorer knowledge of the genealogical information of the individuals presenting such coats, and may derive from the centurial banning of the same and from the genetic rarity. This, together with their longer generation intervals, may have acted as a preserving element of diversity through the avoidance of the increase of inbreeding likely to occur when the time to select offspring selected for breeding is shorter.

The use of breeds with dissimilar genetic backgrounds for specific features such as coat colour, as occurs in PRE and PRá breeds, enhances the opportunities for the maximization of the genetic diversity of the gene pool of the resulting composite breed. Contextually, inbreeding may be avoided when founding breeds do not contribute equally [10]. As a result, despite the likely loss of heterozygosity occurring between the first and second generations [67], further loss of heterozygosity is prevented, which may have contributed to the preservation of a wider variety of coat colour patterns and occurrence of heterozygotically regulated phenotypes in Há horses when compared to PRE and PRá horses, as suggested by the low levels reported for the rate of loss heterozygosity due to inbreeding per generation (Table 6).

Genetic erosion may have occurred at a very low rate. Selection pressure against certain coats, such as diluted coats, may have resulted into slightly higher levels of genetic diversity loss. These values are indicative of a relatively high fraction of diversity being lost in current times, as suggested by other authors. For instance, Fages, et al. [68] reported average loss of over 16% among modern breeds of horses when compared to their 5 millennia old horse ancestors. Furthermore, even relatively reduced levels of inbreeding have been reported for such coat colour subgroups; these inbreeding values may be underrated as a direct consequence of the slightly reduced PCI levels in generations further than the third. Timely action must be taken to control the increase in inbreeding throughout the years across coat colours as levels are starting to approach compromising levels, which may result in the expression of deleterious effects derived from inbreeding in the population, which in the case of coat colours may brought about the appearance of coat colour-linked diseases or conditions [69,70].

Although the under-cover use of diverse mating animals may have contributed to the stabilization of the individual increase in inbreeding across coat colour groups, the presence of highly inbred animals may imply the excessive use of certain individuals still occurs, as has been reported for equines and canines in which the value of animals is related to the value of their ancestors [21]. These findings support the fact that currently the major concern in managing the genetic diversity is the short-term decrease in genetic variability due to the loss of genetic contributions from founders and ancestors, more than the long-term effect of inbreeding itself, as suggested in the literature [71,72,73,74,75,76,77].

The Genetic Conservation Index (GCI) can help to determine the contribution of founders of each coat colour subgroup. McManus, et al. [78] described that GCI computes the genetic contributions of all the identified founders. For this reason, it has been assumed that the animals with higher values of GCI also gather wider fractions of the gene pool of the founding population. For instance, Table 3 supports the historic fact that a forced trend not to register diluted PRE coat ancestors has occurred along history. This translated into GCI values in diluted coats such as isabelline/isabella, pearl and cremello being about half the values in other saturated coats.

Higher than 1–3% inbreeding rates per generation [79] fix deleterious recessive genes at such a speed that selection cannot counteract their effects, which translates in a decrease of population vitality and reproducibility. Inbreeding rates per generation lower than 1% are indicative of populations being partially purged of deleterious genes, which, in turn, maximizes the capacity of populations to tolerate higher rates of inbreeding that may have not been accounted for; for instance, when pedigrees do not reach completeness levels over 75%. In endangered autochthonous populations, such as the Há breed, as stated by the Spanish Official Catalogue of Livestock Breeds [80], a rather conservative approach is recommended.

The number of equivalent subpopulations below 1 revealed a high level of population structuration at a breed level. According to Iglesias Pastrana, et al. [18], population subdivision may be beneficial provided the extinction risk derived from compromising events such as accidents or health-related factors may only cause the disappearance of population sections. Furthermore, genetic diversity may reach its highest levels when populations subdivide into as many separate groups as possible, as occurs when breeding considers coat patterns as a selection criterion [81]. Still, caution should be taken, provided the benefits of subdivision may be counteracted by the negative effects derived from the reduction in effective size and increase in inbreeding.

Effective population size based on the increase in coancestry computation has been deemed to be more biased, but more accurate, than other computational possibilities [82]. These enhanced properties may make effective size estimates based on coancestry more variable and sensitive to the source of information and the data structure considered [83]. Furthermore, De La Rosa, et al. [42] suggested the information provided by this parameter reports effective population size (via individual increase in inbreeding) of a population under random mating.

Additionally, the ratio between both effective population size computations provides information on the degree of population structure [36]. The effective population size computed via the individual increase in inbreeding refers to the effective population size assuming that the partial genetic structure of the population conditioning the mating design is maintained in the future, while the effective population size based on the increase in coancestry assumes that random mating will occur in the near future. This occurred in our population as suggested by the positive levels of non-random mating degree, except for smokey creams for which a non-random mating system where mates are chosen based on dissimilarity of phenotypes may be applied. Given that the difference between both effective population sizes is narrow, it shows how the different colours are, actually, fairly mixed. In fact, considering the certain degree of geographical isolation of the farms involved in the pedigree [84,85], with very little genetic material interchanged by artificial insemination [10], the ratio of both effective population sizes indicates a certain trend to practice inbred mating, focused on maintaining particular colours depending on the owners tastes, which was also denoted for the increase in non-random mating degree levels. Contextually, the subdivision index derived from this practice does not reach too high levels in the current population, as suggested by the low equivalent subpopulations numbers [36].

Although increases in inbreeding may be produced as a response of the population to several factors and conditions, such as population structuring, increases in coancestry reflect the drift caused by the finite size of the population [86]. Consequently, discrepancies between increases in inbreeding and coancestry could be interpreted as cryptic population subdivision [36]. Minimum coancestry mating system, that is mating the individuals in a manner that yields the lower average pairwise coancestry between couples, may lead to lower increases in inbreeding than in coancestry that would be expected under random mating [87].

Breeding policies, breeders or association seeking specific breeding goals, or the relative geographical isolation of herds located at a considerable distance from the rest, may prevent mating between animals for which minimized coancestry levels exist. This in turn may lead to population substructures. Contextually, nonrandom matings may occur as reproduction may mainly take place between individuals within subpopulations [86]. For instance, the particular framework depicted for PRE horses during the period of banning may have responded to this particular promotion of matings seeking to obtain animals displaying bay or grey coat colours. The role of the grey coat, together with the lack of particular knowledge on the genetic background behind coat colours, may have contributed to the palliation of strong substructuring processes, which may support the similar levels for the individual increase in inbreeding found across coat colour subgroups (Table 3). Although substructuring may remarkably increase inbreeding coefficients while coancestries remain approximately stable, it is not easy to confirm if population structuring may be present.

After the study of coancestry matrices, Cervantes [88] concluded that Há individuals share 6.4% of their genes with Arabian horses, with the percentage of genes shared with PRE horses being lower. This had also been reported by Marín Navas, et al. [10], who determined the contribution of PRA to Há horse breed to be of 70.45%. As a result, and provided the presence of certain diluted coat colour patterns which have not been described in the PRá breed, the genetic contribution of PRE horses may be responsible for the regulation of diluted coat colour expression within its 29.55% contribution.

In this regard, the fact that breeders seek particular phenotype characteristics may increase the probability that two individuals share common recent ancestors; hence, coat colours may lay the basis for the development of population substructures as certain coat colour subgroups may, even temporarily, have acted as a breeding nucleus, never receiving alleles from the rest of coat colour subgroup. This may have translated in them having higher increases in inbreeding than the others, while the mean coancestry may have grown equally among all of them [89]. For instance, population substructuring may have conditioned diversity levels when, even if all coat colours except for bay or grey were banned for 32 years, the mating between bay or grey and non-bay or grey individuals may have produced offspring that were excluded from the permitted bay or grey group.

Although in well-known pedigrees, increase in inbreeding and the increase in coancestry converge to an asymptotic value, this only occurs if population subdivision has been permanent along the history of conformation of a certain breed. The sociopolitical and fashion trends conditioning the representativity of each coat colour subgroup in the population may have altered this equilibrium after the years; thus, the reduced effective population size based on the increase in coancestry in comparison to effective population size based on the increase in inbreeding.

Equivalent subpopulation number, or the ratio between effective population size based on the increase in coancestry and based on the increase in inbreeding, measures the degree of population structure [90]. These measures reliably quantify historically cumulated drift and are asymptotically equivalent in an idealized population. Hence, the disagreement between them characterizes the repercussions of preferential matings. A significant deviation from 1 may reflect the presence of a particular management within specific coat colour subgroups because of the existence of diffused subpopulations due to the political and sociological context of a specific breed.

Similar values for both the effective sizes computed via inbreeding and coancestry increase (ratio close to 1) may be indicative of a shallow pedigree. However, the most incomplete coat colour subgroup is that comprising Pearl individuals, with 4.41 equivalent discrete generations and PCI of almost 40% in the 5th generation in the current population. Hence, pedigree information could not be considered incomplete for any of the coat colour subgroups, which is also supported by the maximum values of 0.40 equivalent subpopulations for the Historic Pearl population. The number of equivalent subpopulations was higher in coat colour subgroups which had been banned. According to Cervantes [88], in composite populations (and coat colour subgroups here could be considered as such) the closer the number of equivalent subpopulations is to 1 (even surpassing it), the more dissimilar the structure of originating coat colour subgroups will be. This is obvious because larger numbers of equivalent subpopulations were reported for diluted coat colours (Table 6). Furthermore, the same author suggested that reduced effective population size values are likely to be a consequence of the smaller proportion of matings between individuals of the same coat colour subgroup.

Sørensen, et al. [91] demonstrated that the comparison of *f_e_* and *f_a_* could be used to assess the occurrence of changes in genetic drift and recent bottlenecks in a population, which in the present case is corroborated by the values of *f_e_/f_a_* < 1, respectively. This finding is indicative of the fact that genetic drift may have been stable in the three horse breeds across the coat colour subgroups studied, with a maintained representativity of founders for each of them (Table 4) [91,92]. These values for *f_e_/f_a_,* are concomitant with the increasing trend described by GCI over time, which has only slightly decreased in diluted coat colors. This indicative of the fact that founder representation may have been maintained in all coat colour subgroups except for isabelline, cremello and pearl, which may reflect that a comeback of breeder tastes to the formerly officially recognised coat colours (grey and bay), or popular coat colours such as chestnut/sorrel, may be currently occurring, as suggested by Figure 1, revealing a cyclic pattern.

## 5. Conclusions

Coat colour may be a representative visual trace of the contribution of Spanish Purebred ancestral breed to Hispano-Arabian horses. The lack of knowledge, or lack of consideration of coat colour inheritance patterns, by the entities in charge of individual registration, and the dodging practices of breeders towards the historic banning policies, may have played an important role in the prevention of diversity loss, which is still high in the context of modern horse breeds. Hispano-Arabian crossbred horse breeding may increase the likelihood of double dilute combinations occurring, as denoted by the increased number of Hispano-Arabian horses displaying diluted coats such as pearl. However, the historic ban occurring against them may have promoted the lack of genealogical information in generations from the fourth generation on, and the lengthened generation intervals. The differential consideration of rather complete genealogies for the calculation of genetic diversity parameters such as inbreeding, may have contributed to the greater or lesser accuracy of their estimation across coat colour possibilities. A continuously evolving diversity panorama is depicted regarding how the different coats have been managed and bred for. However, a cyclic comeback of breeder taste to the formerly officially recognised coat colours (grey and bay) and chestnut/sorrel, may be occurring currently.

## Figures and Tables

**Figure 1 vetsci-09-00068-f001:**
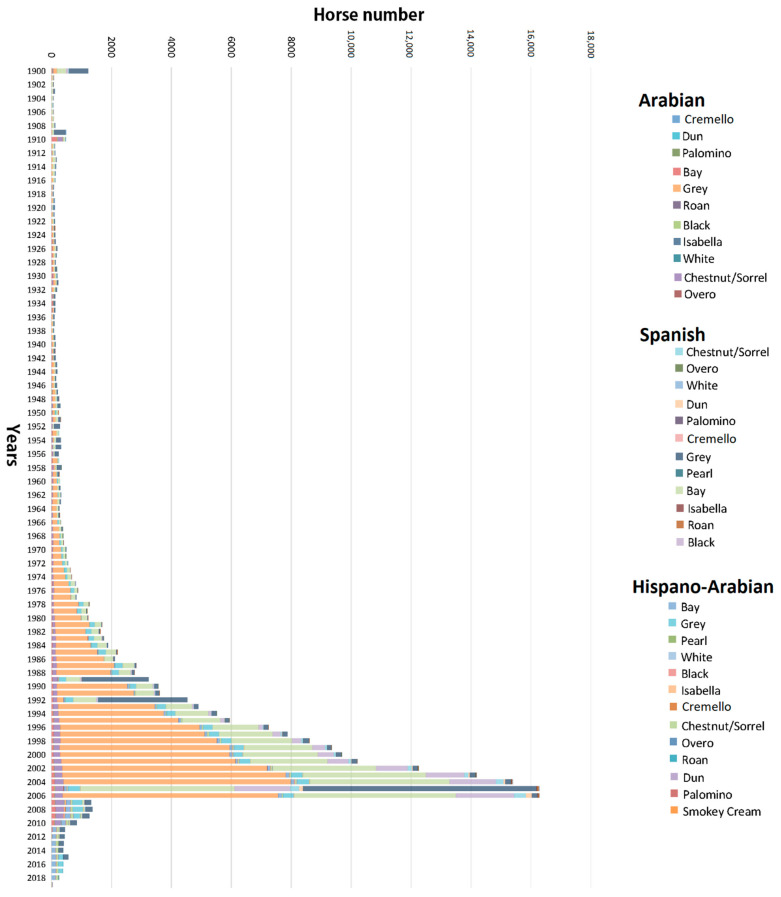
Evolution of birth number and coat distribution across breeds in the historic population of Arabian and Spanish Purebred and Hispano-Arabian horses from 1900 to 2019.

**Figure 2 vetsci-09-00068-f002:**
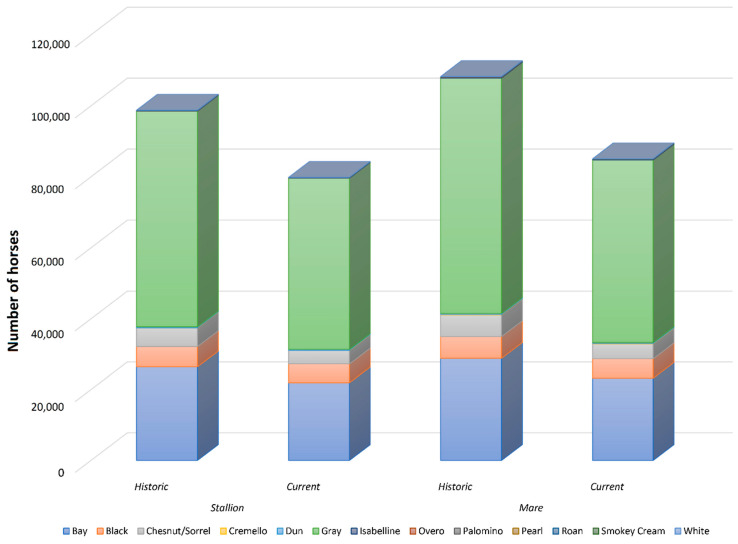
Coat colour subgroups, sample size historic and current distribution. Coat colours are displayed in increasing order from left to right in the legend, and from bottom to top in graphic form depending on their relative frequency.

**Figure 3 vetsci-09-00068-f003:**
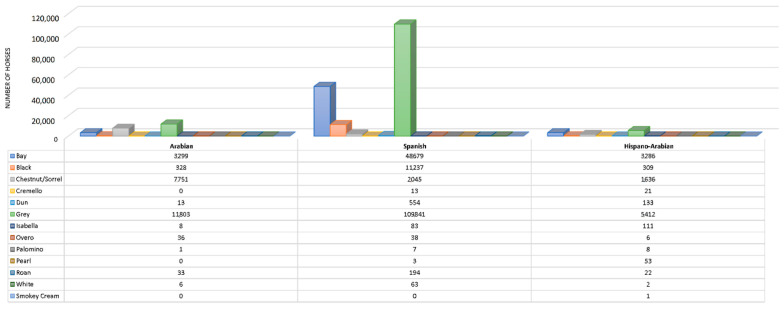
Coat colour subgroups sample size distribution across breeds (Spanish and Arabian Purebred and Hispano-Arabian horse breeds). Colours listed in table legend from top to bottom are presented in graphic from left to right.

**Figure 4 vetsci-09-00068-f004:**
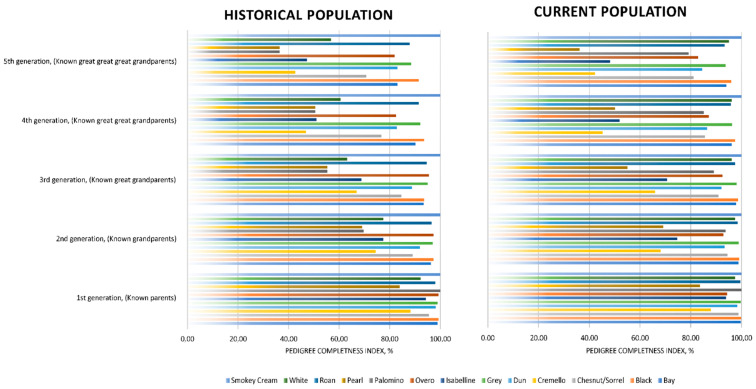
Historic and current pedigree completeness indices (1st, 2nd, 3rd, 4th and 5th generation) across coat colour subgroups.

**Figure 5 vetsci-09-00068-f005:**
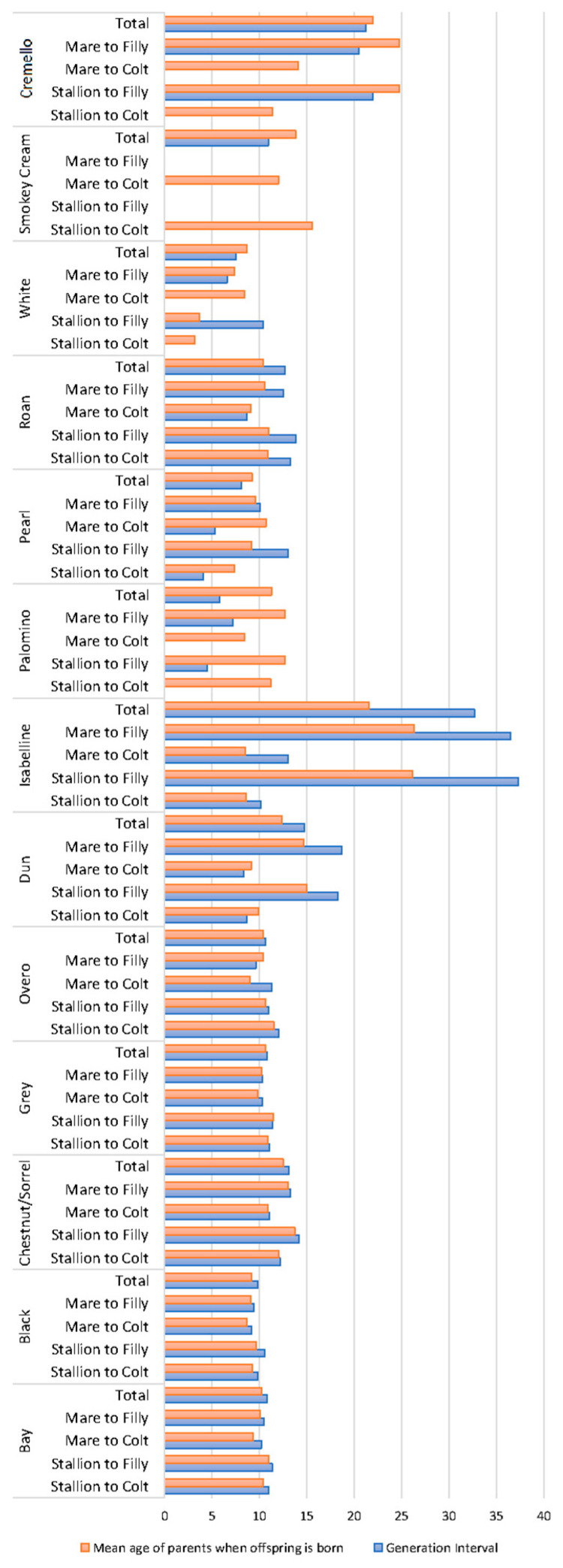
Average generational intervals and mean age of the parents at the birth of their offspring (years) across coat colour subgroups.

**Figure 6 vetsci-09-00068-f006:**
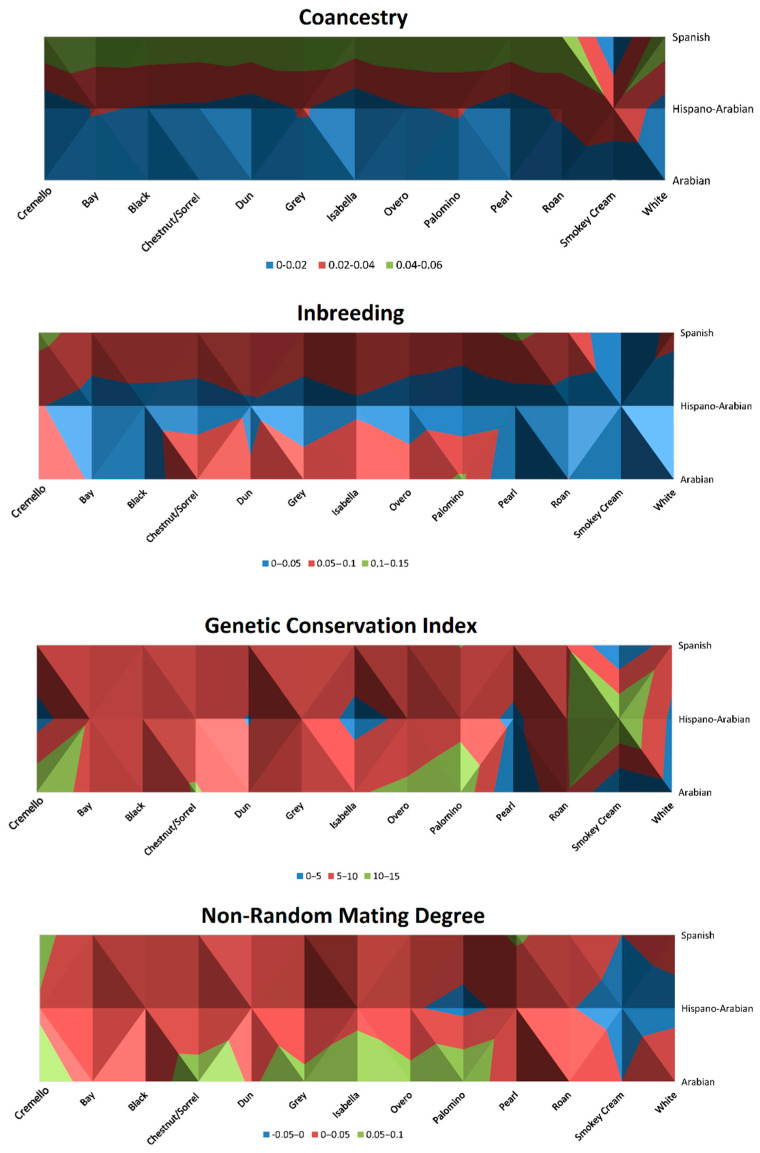
Distribution of nonrandom mating degree (*α*), inbreeding rate (*F*), coancestry (C) and Genetic Conservation Index (GCI) across coat colour subgroups for the Arabian and Spanish Purebred and Hispano-Arabian horses. The graphic should be interpreted as a bird’s-eye view.

**Figure 7 vetsci-09-00068-f007:**
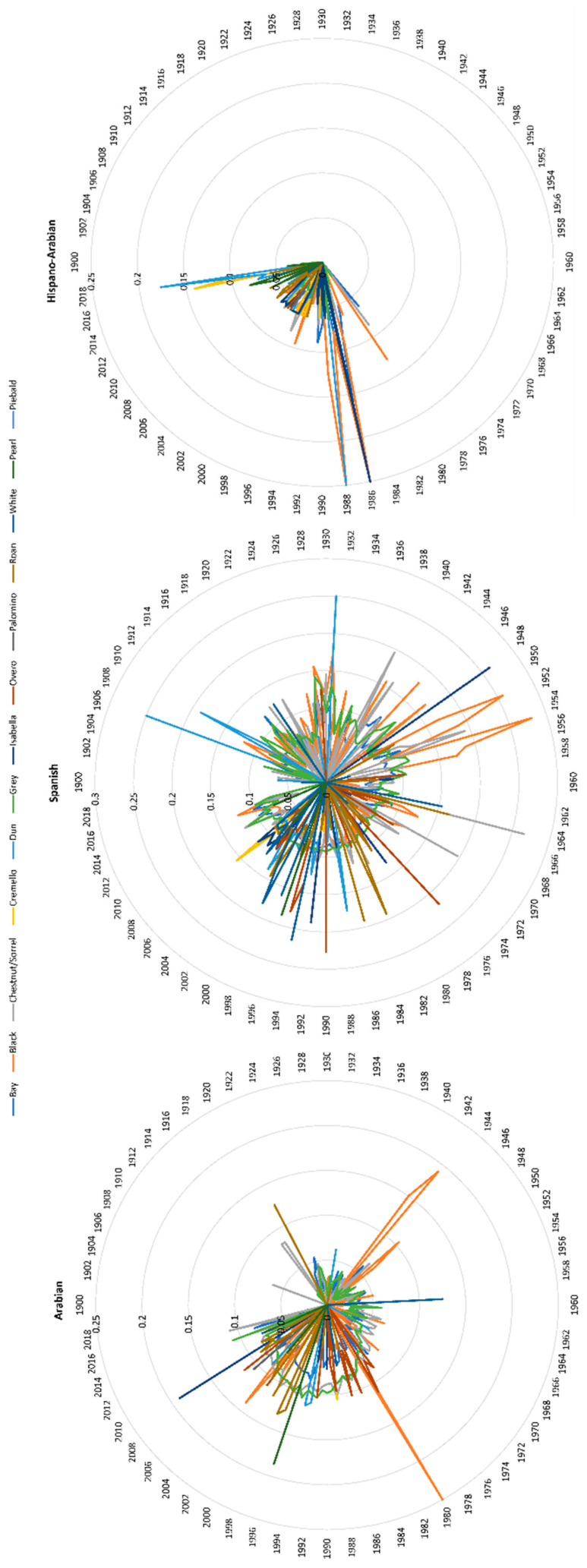
Biyearly evolution of average inbreeding (*F*) from 1900 to 2019 across coat colours subgroups in the Arabian (PRá) and Spanish Purebred (PRE) and Hispano-Arabian (Há) horse breeds.

**Figure 8 vetsci-09-00068-f008:**
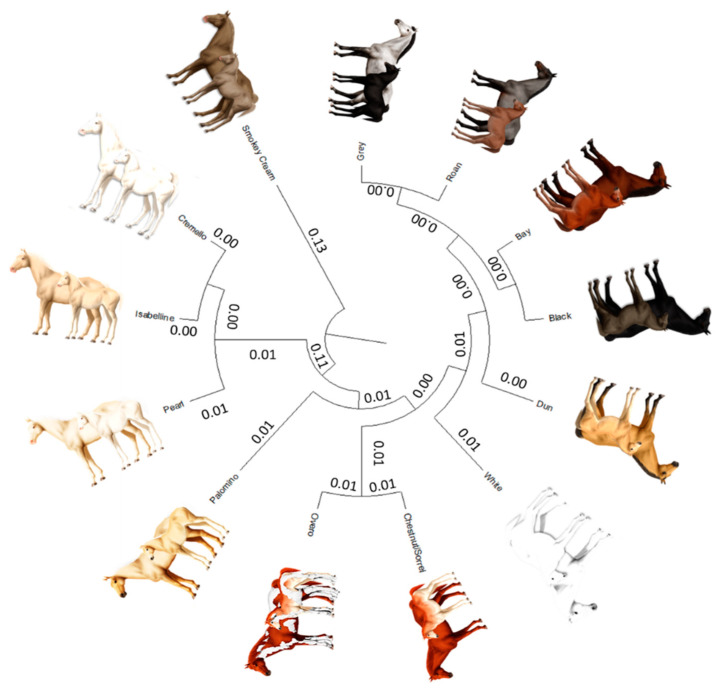
Cladogram constructed from Nei’s genetic distances among coat colour subgroups.

**Table 1 vetsci-09-00068-t001:** Summary of historic and current sample demographics.

Set	Breed	Source	Number of Animals	Males/Females	Period (Birthdates)
Historic	Hispano-Arabian	Spanish Union of Purebred Hispano-Arabian Horses Breeders (UEGHá)	11,010	4268 males and 6742 females	January 1950 and April 2019
	Spanish Purebred	National Association of Purebred Spanish Horse Breeders (ANCCE)	172,797	83,408 males and 89,389 females	January 1884 and July 2019
	Arabian Purebred	Spanish Association of Arabian Horse Breeders (AECCA)	23,293	11,143 males and 12,150 females	January 1898 and June 2019
Current	Hispano-Arabian	Spanish Union of Purebred Hispano-Arabian Horses Breeders (UEGHá)	9997	4031 males and 5966 females	December 1984 and April 2019
	Spanish Purebred	National Association of Purebred Spanish Horse Breeders (ANCCE)	141,357	69,184 males and 72,163 females	April 1984 and July 2019
	Arabian Purebred	Spanish Association of Arabian Horse Breeders (AECCA)	13,576	6632 males and 6944 females	June 1985 and June 2019

**Table 2 vetsci-09-00068-t002:** Summary of results of generation analysis across coat colour population subgroups.

Parameter	Maximum Number of Traced Generations, n		Number of Maximum Generations (Mean ± SD)		Number of Complete Generations (Mean ± SD)		Number of Equivalent Generations (Mean ± SD)	
Population Set	Historic	Current	Historic	Current	Historic	Current	Historic	Current
Grey	20.00	20.00	13.67 ± 3.47	14.59 ± 3.46	4.36 ± 1.43	4.65 ± 1.42	7.98 ± 2.09	8.53 ± 2.09
Bay	21.00	21.00	14.01 ± 3.47	15.11 ± 3.46	4.44 ± 1.43	4.79 ± 1.42	8.03 ± 2.09	8.69 ± 2.09
Chestnut/Sorrel	20.00	20.00	11.66 ± 3.47	13.54 ± 3.46	3.66 ± 1.43	4.12 ± 1.42	6.51 ± 2.09	7.51 ± 2.09
Black	20.00	20.00	15.01 ± 3.46	15.66 ± 3.46	4.81 ± 1.43	5.01 ± 1.42	8.71 ± 2.09	9.09 ± 2.09
Overo	17.00	17.00	11.24 ± 3.34	13.29 ± 3.21	3.73 ± 1.40	4.23 ± 1.38	6.57 ± 2.02	7.79 ± 1.95
Roan	19.00	19.00	14.48 ± 3.44	15.42 ± 3.54	4.51 ± 1.42	4.78 ± 1.44	8.29 ± 2.08	8.84 ± 2.12
Dun	19.00	19.00	14.14 ± 3.43	14.74 ± 3.47	4.33 ± 1.42	4.51 ± 1.42	8.00 ± 2.07	8.34 ± 2.08
White	18.00	18.00	10.17 ± 3.43	15.36 ± 3.64	3.32 ± 1.42	5.08 ± 1.47	5.94 ± 2.07	8.92 ± 2.18
Isabelline	18.00	18.00	9.74 ± 3.42	9.97 ± 3.48	2.93 ± 1.42	3.03 ± 1.43	5.36 ± 2.06	5.52 ± 2.09
Cremello	18.00	18.00	9.21 ± 3.46	9.15 ± 3.46	2.88 ± 1.42	2.82 ± 1.42	5.16 ± 2.07	5.10 ± 2.07
Pearl	19.00	19.00	13.09 ± 3.72	13.07 ± 1.08	1.63 ± 1.49	1.62 ± 1.49	4.44 ± 2.23	4.42 ± 2.23
Palomino	19.00	19.00	15.31 ± 4.23	15.31 ± 4.23	3.94 ± 1.67	3.94 ± 1.67	8.05 ± 2.53	8.05 ± 2.53
Smokey Cream	18.00	18.00	18.00 ± 0.00	18.00 ± 0.00	5.00 ± 0.00	5.00 ± 0.00	5.00 ± 0.00	9.61 ± 0.00

**Table 3 vetsci-09-00068-t003:** Summary of demographic/population statistics derived from the analysis of the pedigree across coat colour subgroups.

Coat Colour Subgroup	Population Sets	Inbreeding Coefficient (F), (%)	Individual Increase of Mean Inbreeding (ΔF), (%)	Maximum Inbreeding Coefficient %	Inbred Animals, %	Highly Inbred Animals, %	Average Coancestry (C), %	Average Relatedness Coefficient (ΔR), %	Non-Random Mating Rate (α)	GCI
Grey	Historic	8.33	1.12	55.04	18.41	26.28	5.08	10.15	0.03	9.37
	Current	8.52	1.06	55.04	16.90	26.07	5.23	10.47	0.03	9.81
Bay	Historic	7.38	1.03	53.91	31.12	18.65	4.85	9.69	0.03	9.25
	Current	7.56	0.94	49.61	32.10	17.86	5.02	10.04	0.03	9.83
Chestnut/Sorrel	Historic	6.95	1.10	46.88	75.78	26.75	1.64	3.28	0.05	9.74
	Current	7.76	1.12	46.88	81.66	29.83	1.90	3.80	0.06	10.73
Black	Historic	7.54	0.96	45.31	95.27	16.64	5.18	10.36	0.03	9.66
	Current	7.64	0.90	43.38	98.01	16.21	5.26	10.51	0.03	10.00
Overo	Historic	7.81	1.22	22.72	90.00	31.25	3.55	6.42	0.05	9.59
	Current	7.31	0.98	17.89	94.29	25.71	3.55	7.10	0.04	9.95
Roan	Historic	7.59	0.95	28.62	91.16	21.29	4.71	9.43	0.03	9.61
	Current	7.87	0.93	28.62	93.90	20.19	4.96	9.91	0.03	10.03
Dun	Historic	7.19	1.12	32.42	89.43	14.00	4.68	9.35	0.03	8.69
	Current	7.41	1.09	32.42	92.72	14.24	4.76	9.52	0.03	8.92
White	Historic	6.25	1.24	25.00	64.79	21.13	4.86	9.71	0.01	6.98
	Current	8.80	1.01	21.52	94.87	28.21	5.31	10.62	0.04	9.86
Isabelline	Historic	6.78	1.68	37.19	7.67	1.49	2.61	5.21	0.04	5.85
	Current	6.96	1.67	37.19	80.77	14.84	2.63	5.25	0.04	5.83
Cremello	Historic	7.41	1.86	23.67	89.24	20.59	2.33	4.67	0.05	5.68
	Current	7.32	1.87	23.67	87.88	18.18	2.38	4.75	0.05	5.45
Pearl	Historic	2.86	0.60	22.66	30.26	8.93	1.20	2.39	0.02	4.06
	Current	2.91	0.61	22.66	30.91	9.09	1.16	2.33	0.02	4.04
Palomino	Historic	4.75	0.70	11.10	62.50	12.50	3.69	7.38	0.01	9.44
	Current	4.75	0.70	11.10	62.50	12.50	3.69	7.38	0.01	9.44
Smokey Cream	Historic	1.81	0.20	1.81	1.00	0.00	3.65	7.30	−0.02	15.07
	Current	1.81	0.21	1.81	1.00	0.00	3.65	7.30	−0.02	15.07

**Table 4 vetsci-09-00068-t004:** Analysis of gene origin across coat colour population subgroups.

Coat Colour Subgroup	Population Set	Base Population (One or More Unknown Parents)	Actual Base Population (One Unknown Parent = Half Founder)	Number of Founders, n	Number of Ancestors, n	Effective Number of Non-Founders (*N_ef_*)	Number of Founder Equivalents (*f_e_*)	Effective Number of Ancestors (*f_a_*)	Founder Genome Equivalents (*f_g_*)	*f_a_/f_e_* Ratio	*f_g_/f_e_* Ratio
Grey	Historic	1503.00	331.00	1172.00	2144.00	14.66	26.50	17.00	9.23	0.64	0.35
	Current	277.00	49.00	228.00	1138.00	12.99	26.03	17.00	8.67	0.65	0.33
Bay	Historic	600.00	84.00	516.00	1490.00	16.14	25.14	17.00	9.83	0.68	0.39
	Current	128.00	28.00	100.00	840.00	14.31	25.68	17.00	9.19	0.66	0.36
Chestnut/Sorrel	Historic	558.00	88.00	470.00	1283.00	20.26	58.99	31.00	15.08	0.53	0.26
	Current	108.00	22.00	86.00	762.00	17.44	58.61	27.00	13.44	0.46	0.23
Black	Historic	84.00	22.00	62.00	500.00	13.67	22.41	15.00	8.49	0.67	0.38
	Current	23.00	3.00	20.00	383.00	12.74	23.25	15.00	8.23	0.65	0.35
Overo	Historic	4.00	0.00	4.00	106.00	16.89	51.02	29.00	12.69	0.57	0.25
	Current	2.00	0.00	2.00	52.00	14.04	45.66	26.00	10.74	0.57	0.24
Roan	Historic	5.00	0.00	5.00	195.00	15.38	31.39	20.00	10.32	0.64	0.33
	Current	1.00	0.00	1.00	177.00	13.70	28.97	19.00	9.30	0.66	0.32
Dun	Historic	14.00	2.00	12.00	210.00	13.30	27.76	18.00	8.99	0.65	0.32
	Current	31.00	21.00	10.00	191.00	12.21	28.60	18.00	8.56	0.63	0.30
White	Historic	5.00	0.00	5.00	107.00	17.72	13.55	10.00	7.68	0.64	0.35
	Current	1.00	0.00	1.00	69.00	9.89	23.10	15.00	6.93	0.65	0.33
Isabelline	Historic	11.00	0.00	11.00	138.00	12.51	26.40	12.00	8.49	0.68	0.39
	Current	11.00	0.00	11.00	129.00	10.52	27.46	10.00	7.61	0.66	0.36
Cremello	Historic	4.00	0.00	4.00	27.00	8.79	28.17	9.00	6.70	0.53	0.26
	Current	4.00	0.00	4.00	26.00	8.37	26.66	9.00	6.37	0.46	0.23
Pearl	Historic	9.00	0.00	9.00	39.00	14.53	32.87	16.00	10.08	0.67	0.38
	Current	9.00	0.00	9.00	38.00	14.67	32.83	17.00	10.14	0.65	0.35
Palomino	Historic	0.00	0.00	0.00	19.00	8.85	42.81	17.00	7.33	0.57	0.25
	Current	0.00	0.00	0.00	19.00	8.85	42.51	17.00	7.33	0.57	0.24
Smokey Cream	Historic	0.00	0.00	0.00	1.00	1.01	37.35	1.00	0.98	0.46	0.23
	Current	0.00	0.00	0.00	1.00	0.00	0.00	1.00	0.00	0.39	0.19

**Table 5 vetsci-09-00068-t005:** Measures of genetic diversity and genetic diversity loss across coat colour subgroup.

Coat Colour Subgroup	Population Set	Genetic Diversity GD (%)	Genetic Diversity Loss GDL (%)	GDL Due to Genetic Drift since Founders (%)	GDL Due to Bottlenecks and Genetic Drift since Founders (GL) (%)	GDL Due to Unequal Founder Contributions (%)	Ancestors Explaining 25% of the Gene Pool (n)	Ancestors Explaining 50% of the Gene Pool (n)	Ancestors Explaining 75% of the Gene Pool (n)
Grey	Historic	95	5	2	4	5	2	6	20
	Current	94	6	2	4	6	2	6	18
Bay	Historic	95	5	2	3	5	2	6	22
	Current	95	5	2	3	5	2	6	20
Chestnut/Sorrel	Historic	97	3	1	2	3	3	12	48
	Current	96	4	1	3	4	3	11	35
Black	Historic	94	6	2	4	6	2	6	17
	Current	94	6	2	4	6	2	6	17
Overo	Historic	96	4	1	3	4	4	11	28
	Current	95	5	1	4	5	4	10	23
Roan	Historic	95	5	2	3	5	3	8	26
	Current	95	5	2	4	5	3	7	22
Dun	Historic	94	6	2	4	6	2	7	17
	Current	94	6	2	4	6	3	7	17
White	Historic	93	7	4	3	7	2	4	16
	Current	93	7	2	5	7	2	6	15
Isabelline	Historic	94	6	2	4	6	2	4	16
	Current	93	7	2	5	7	2	4	15
Cremello	Historic	93	7	2	6	7	2	4	12
	Current	92	8	2	6	8	2	4	11
Pearl	Historic	95	5	2	3	5	2	7	18
	Current	95	5	2	3	5	2	7	18
Palomino	Historic	93	7	1	6	7	3	7	12
	Current	93	7	1	6	7	3	7	12
Smokey Cream	Historic	49	51	1	50	51	1	1	1
	Current	0	0	N/A	N/A	N/A	1	1	1

N/A: Computation not feasible. Only one animal present.

**Table 6 vetsci-09-00068-t006:** Summary of results for effective population size calculated from the individual inbreeding rate and by the individual coancestry rate and the number of equivalent subpopulations.

	Parameter	Population Set	Effective Population Size Calculated through of the Individual Inbreeding Rate	Effective Size of the Population Calculated through the Individual Coancestry Rate	Number of Equivalent Subpopulations	Rate of Loss Heterozygosity Due to Inbreeding per Generation	
Coat Colour Subgroup							
Grey		Historic	96.15	16.13	0.17	0.005	
		Current	138.89	10.06	0.07	0.004	
Bay		Historic	44.64	4.93	0.11	0.011	
		Current	47.17	4.78	0.10	0.011	
Chestnut/Sorrel		Historic	48.54	5.16	0.11	0.010	
		Current	53.19	4.98	0.09	0.009	
Black		Historic	45.45	15.24	0.34	0.011	
		Current	44.64	13.16	0.29	0.011	
Overo		Historic	52.08	4.83	0.09	0.010	
		Current	55.56	4.76	0.09	0.009	
Roan		Historic	40.98	7.79	0.19	0.012	
		Current	51.02	7.04	0.14	0.010	
Dun		Historic	60.98	6.35	0.10	0.008	
		Current	75.76	6.35	0.08	0.007	
White		Historic	44.64	5.35	0.12	0.011	
		Current	45.87	5.25	0.11	0.011	
Isabelline		Historic	40.32	5.15	0.13	0.012	
		Current	49.50	4.71	0.10	0.010	
Cremello		Historic	29.76	9.60	0.32	0.017	
		Current	29.94	9.52	0.32	0.017	
Pearl		Historic	26.88	10.71	0.40	0.019	
		Current	26.74	10.53	0.39	0.019	
Palomino		Historic	83.33	20.92	0.25	0.006	
		Current	81.97	21.46	0.26	0.006	
Smokey Cream		Historic	N/A	N/A	N/A	N/A	
		Current	N/A	N/A	N/A	N/A	

N/A: Computation not feasible. Only one animal present.

**Table 7 vetsci-09-00068-t007:** Results for Wright’s F fixation or statistical indices, F_IS_ (coefficient of inbreeding relative to subpopulation), F_ST_ (correlation between random gametes extracted from subpopulation relative to total population) and F_IT_ (coefficient of inbreeding relative to total population).

Parameters	Coat Colour Subgroup	
Populational Set	Historic	Current
F_IS_ (Inbreeding coefficient relative to the subpopulation)	0.029	0.018
F_ST_ (Correlation between random gametes drawn from the subpopulation relative to the total population)	0.004	0.014
F_IT_ (Inbreeding coefficient relative to the total population)	0.033	0.032
Mean inbreeding within subpopulations	0.079	0.057
Mean number of horses per subpopulation	12,943.75	10,308.81
Number of Nei genetic distances	78	78
Average Nei genetic distance	0.004	0.014
Mean coancestry within subpopulations	0.052	0.040
Self coancestry	0.540	0.528
Mean coancestry in the metapopulation	0.048	0.026
Subpopulations	13	13

## Data Availability

Data will be made available from corresponding author upon reasonable request.

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
