# Peer review of "One Hundred Years of Coat Colour Influences on Genetic Diversity in the Process of Development of a Composite Horse Breed"

_vetsci, 2022, doi:10.3390/vetsci9020068_

Round 1

Reviewer 1 Report

Dear Authors, please correct the marked sentences in conclusion (attached file). Could you include the inbreeding numbers into the abstract.

The research main addressed the inbreeding factor in different Spanish breeds and the performance associated with the coat colour of breeding horses.

Based on the inbreeding factor and the genetic tests the horse owner or veterinarian can decide which stallion is the best for breeding.

Compared with other published material, this research has investigated the period from 1898 to 2019 and therefore originated.

Regarding the methodology, the authors consider could consider improvements the evaluation of inbreeding (2.3.) and the probabilities of gene origin (2.4.).

Based on the results the conclusion and the abstract could be better with evaluated inbreeding factors of different coat colours.

The references are appropriate.

On the tables and figures, Figure 6 is only understandable for geneticists.

Author Response

Reviewer 1

Dear Authors, please correct the marked sentences in conclusion (attached file). Could you include the inbreeding numbers into the abstract.

Response: Corrected. Inbreeding numbers were added to the abstract as requested.

The research main addressed the inbreeding factor in different Spanish breeds and the performance associated with the coat colour of breeding horses.

Based on the inbreeding factor and the genetic tests the horse owner or veterinarian can decide which stallion is the best for breeding.

Response: We agree with the reviewer and added this aspect to the body text of the manuscript.

Compared with other published material, this research has investigated the period from 1898 to 2019 and therefore originated.

Response: Thank you for the appreciation. We appreciate reviewer kind comment.

Regarding the methodology, the authors consider could consider improvements the evaluation of inbreeding (2.3.) and the probabilities of gene origin (2.4.).

Based on the results the conclusion and the abstract could be better with evaluated inbreeding factors of different coat colours.

Response: We followed the reviewer suggestion.

The references are appropriate.

Response: We thank the reviewer for his/her kind comments.

On the tables and figures, Figure 6 is only understandable for geneticists.

Response: We understand the reviewer concern. However, this figure is important as it is given it provides an overall picture of the situation of important diversity parameters. As we understand it may be complex to interpret for nonspecific researchers, we also provided this as a rather extense information in Table 6.

Reviewer 2 Report

1) I find your manuscript difficult to read as the use of English language leaves much room for improvement. I did set out to produce a marked up version of the document for you but in the end had to give up as it was becoming a much bigger task than I have time to devote to. It needs revising by someone who understands plain English and does not choose inappropriate words and synonyms from a thesaurus. Even the title of your paper is confusing as few will understand the meaning of the word Centurial. You might be better to change this to "Historic coat colour influences" or if you feel you need to refer to a time frame "One hundred years of coat colour influences".

2) Your line numbering bizarrely  only starts on page 9 of the manuscript after Figure 3. This makes it difficult to refer to specific lines in your text that I feel would benefit revision.

3) In your Abstract  you need to revise your choice of words  in these 3 phrases "knowledge disconsideration" ; "during decades" & "parallelly evolve with".

4) In Page 2 "would share thematic with" ; "conferred to phantroptical features" both need putting into plain English.

5) On Page 3 "mention to coat colour" should read "mention of coat colour"

6) Also on Page 3 "heterosis complementarity" ; "prioritary" ; "unbalancedly" ; "cluelessness" ( perhaps you mean "lack of understanding") ; "occurring along the history" (could be replaced with "occurring throughout history); "inheritance mechanisms knowledge" could read "knowledge of the mechanisms of inheritance".

7) Page 4 "Conclusively" could be replaced with "In conclusion"; "along the history" appears again - see comment 6 ;"Genetic analyses were performed considering a population set......" could read "Genetic analysis was conducted on a filtered dataset containing only living animals"

8) Your Figure 1 is very hard to interpret at the published size - it needs either a full page or moving to supplementary material where it can be viewed at full screen resolution. I also noted that your colour legend for the various coat colour groups is not consistent throughout your other Figures. This makes it hard to compare the various graphical representations of your findings.

9) Page 7 "methods in (37)". It would help the reader to mention the citation author in the main body of the text the first time you refer to the work  as "methods described by Cervantes et al (37)"

10) Page 8 Results :- delete "the" in "and the 88.03%"

11) Page 9 Figures 2& 3: again both figures are almost impossible to read or interpret at the small size in which they appear. They warrant a full page each or putting into supplementary material and referring to in the text. A consistent colour legend across all of your relevant Figures/graphics  for the various coat colours would aid interpretation and comparison.

12) Page 9 " being reported by for Cremello"

13) Figure 4 Even at full page the graph is hard to interpret as it comtains so much data. Probably better in Supplementary material where it can be viewed at full screen resolution. Again a consistent colour legend will improve all your relevant figures.

14) Figure 5: - Make whole page or move to supplementary material . A generation interval of 10 years is consistent with many other horse and pony breeds. 20 years is significantly different and 30 years flags alarm bells. To me this is a data quality / mis-recording matter probably best attributed to the banning policies you refer to in your discussion.

15) Figure 6 is very "pretty but I find the style of the graphic makes it difficult to interpret the data. Your paper would be improved if you presented this data in a different way ( and used a consistent colour legend as with other figures)

16) Figure 7 can not be read at this scale - each graph warrants a full page or move to supplementary material. Use a consistent colour scheme across all figures for each coat colour permutation.

17) Line 334  "Contextually a higher repercussion " makes no sense

18) Line 757 "in the palliation of diversity loss" makes no sense

19) You refer to the historic banning policies that have been applied to the various breed standards over the years and how that may have affected the way not only horses have been bred but how they may have been registered with false or incorrect information or not registered at all. This problem is not unique to Spanish horse breeds. It has for example affected the Cleveland Bay Horse studbook where purebred animals that have been born chestnut have historically been registered as part-breds or not registered at all. Nowadays under EU Zootechnical Legislation this practice is forbidden and such animals have to be registered as purebred in the pure bred studbook although they may be described as "colour chestnut does not conform to breed standard".

20) I think it is important to refer to the various methods of evaluating Effective Population Size and you quite correctly do not fall into the trap of reporting Census effective population size based on the number of breeding males and females in the population. Endog (of which I have a lot of experience) is not the only freeware that can carry out these calculations. Have you though of using PopRep (Groeneveld et al). You can run your data on their servers at https://popreport.fli.de/cgi-bin/entry.pl The resulting monitoring report takes you through a staged system of choice for which method of determining Effective Population Size is most appropriate for your data. Translating an Endog input file into PopRep format is not too difficult. It might be worth doing this with your data.

21) If you have access to SNP or microsatellite data (from parentage testing) you could conduct STRUCTURE analysis to see if there really is population substructure linked to coat colour based on DNA evidence. Your paper relies on studbook / pedigree records that by your own admission are flawed and thus far from 100% reliable. Always a problem with studbook records because of human error be that intentional or not. PopRep does have good error checking routines that may pick up things that Endog does not.

22) Lacey et al point out the dangers of a breed management scheme that directly minimised either inbreeding or coancestry in the next generation. Better to manage inbreeding indirectly through managing mean kinship first and foremost for sustainable breeding. The aim is to increase effective population size by slowly reducing the rate of increase in inbreeding. Any sudden decrease in one generation can lead to a rebound in future generations as you increase the frequency of not only rare alleles but the more common ones as well and bring them together in a way that they can not be separated out in later generations. (Oliehoek et al explain this well http://www.breedingfordiversity.com)

Author Response

Reviewer 2

Comments and Suggestions for Authors

1) I find your manuscript difficult to read as the use of English language leaves much room for improvement. I did set out to produce a marked up version of the document for you but in the end had to give up as it was becoming a much bigger task than I have time to devote to. It needs revising by someone who understands plain English and does not choose inappropriate words and synonyms from a thesaurus. Even the title of your paper is confusing as few will understand the meaning of the word Centurial. You might be better to change this to "Historic coat colour influences" or if you feel you need to refer to a time frame "One hundred years of coat colour influences".

Response: We thank the reviewer for his/her attention and work on this manuscript. The paper was revised by a Cambridge ESOL Examination instructor and grammar and typos were sought after to improve readability and reading quality. Title was changed according to reviewer’s suggestions.

2) Your line numbering bizarrely  only starts on page 9 of the manuscript after Figure 3. This makes it difficult to refer to specific lines in your text that I feel would benefit revision.

Response: We apologize for this. We have not noticed about it. We change it and now there is continuous line numbering.

3) In your Abstract  you need to revise your choice of words  in these 3 phrases "knowledge disconsideration" ; "during decades" & "parallelly evolve with".

Response: Suggestions were followed.

4) In Page 2 "would share thematic with" ; "conferred to phantroptical features" both need putting into plain English.

Response: Suggestion were followed. However, we decided to keep the word phaneroptical given it is a common word use in papers of the same nature and it is the best word to describe what we wanted to describe in the paper.

5) On Page 3 "mention to coat colour" should read "mention of coat colour"

Response: Suggestions were followed.

6) Also on Page 3 "heterosis complementarity" ; "prioritary" ; "unbalancedly" ; "cluelessness" ( perhaps you mean "lack of understanding") ; "occurring along the history" (could be replaced with "occurring throughout history); "inheritance mechanisms knowledge" could read "knowledge of the mechanisms of inheritance".

Response: Suggestions were followed.

7) Page 4 "Conclusively" could be replaced with "In conclusion"; "along the history" appears again - see comment 6 ;"Genetic analyses were performed considering a population set......" could read "Genetic analysis was conducted on a filtered dataset containing only living animals"

Response: Suggestions were followed.

8) Your Figure 1 is very hard to interpret at the published size - it needs either a full page or moving to supplementary material where it can be viewed at full screen resolution. I also noted that your colour legend for the various coat colour groups is not consistent throughout your other Figures. This makes it hard to compare the various graphical representations of your findings.

Response: We followed the reviewer suggestion and changed the distribution of the figure for it to fit a single page, as we really think it is important to keep it in the manuscript. The colour issue that the reviewer addressed was indeed chosen as for this figure a different colour was needed for each coat colour group within each breed in particular. This does not occur in the rest of figures, hence the disagreement in colours.

9) Page 7 "methods in (37)". It would help the reader to mention the citation author in the main body of the text the first time you refer to the work  as "methods described by Cervantes et al (37)"

Response: Suggestions was followed.

10) Page 8 Results :- delete "the" in "and the 88.03%"

Response: Suggestions was followed.

11) Page 9 Figures 2& 3: again both figures are almost impossible to read or interpret at the small size in which they appear. They warrant a full page each or putting into supplementary material and referring to in the text. A consistent colour legend across all of your relevant Figures/graphics  for the various coat colours would aid interpretation and comparison.

Response: We followed the reviewer suggestion and changed the distribution of the figures for them to fit a single page. The colour issue that the reviewer addressed corrected and now both figures have the same colour code.

12) Page 9 " being reported by for Cremello"

Response: Corrected.

13) Figure 4 Even at full page the graph is hard to interpret as it comtains so much data. Probably better in Supplementary material where it can be viewed at full screen resolution. Again a consistent colour legend will improve all your relevant figures.

Response: We corrected it. The colour issue that the reviewer addressed corrected and now figures have the same colour code. As the journal is published online. The figure has enough quality as to be enlarged if it is necessary.

14) Figure 5: - Make whole page or move to supplementary material . A generation interval of 10 years is consistent with many other horse and pony breeds. 20 years is significantly different and 30 years flags alarm bells. To me this is a data quality / mis-recording matter probably best attributed to the banning policies you refer to in your discussion.

Response: We corrected it. The colour issue that the reviewer addressed corrected and now figures have the same colour code. As the journal is published online. The figure has enough quality as to be enlarged if it is necessary. We agree with the reviewer, however, these generation intervals appear in diluted coats which frequently appear as a result of recessive genes playing their part as clarified in the body text.

15) Figure 6 is very "pretty but I find the style of the graphic makes it difficult to interpret the data. Your paper would be improved if you presented this data in a different way (and used a consistent colour legend as with other figures).

Response: We understand the reviewer concern. However, this figure is important as it is given it provides an overall picture of the situation of important diversity parameters. As we understand it may be complex to interpret for nonspecific researchers, we also provided this as a rather extense information in Table 6.

16) Figure 7 can not be read at this scale - each graph warrants a full page or move to supplementary material. Use a consistent colour scheme across all figures for each coat colour permutation.

Response: We corrected it. The colour issue that the reviewer addressed corrected and now figures have the same colour code. As the journal is published online. The figure has enough quality as to be enlarged if it is necessary

17) Line 334  "Contextually a higher repercussion " makes no sense

Response: This finding derives from the fact that Há hores can have a up to 75% of PRá blood.

18) Line 757 "in the palliation of diversity loss" makes no sense

Response: Corrected.

19) You refer to the historic banning policies that have been applied to the various breed standards over the years and how that may have affected the way not only horses have been bred but how they may have been registered with false or incorrect information or not registered at all. This problem is not unique to Spanish horse breeds. It has for example affected the Cleveland Bay Horse studbook where purebred animals that have been born chestnut have historically been registered as part-breds or not registered at all. Nowadays under EU Zootechnical Legislation this practice is forbidden and such animals have to be registered as purebred in the pure bred studbook although they may be described as "colour chestnut does not conform to breed standard".

Response: We agree and added this interesting information to our manuscript. However, now all coats are accepted in the Há horse, hence this is not longer a problem, although still misidentification may occur.

20) I think it is important to refer to the various methods of evaluating Effective Population Size and you quite correctly do not fall into the trap of reporting Census effective population size based on the number of breeding males and females in the population. Endog (of which I have a lot of experience) is not the only freeware that can carry out these calculations. Have you though of using PopRep (Groeneveld et al). You can run your data on their servers at https://popreport.fli.de/cgi-bin/entry.pl The resulting monitoring report takes you through a staged system of choice for which method of determining Effective Population Size is most appropriate for your data. Translating an Endog input file into PopRep format is not too difficult. It might be worth doing this with your data.

Response: We do know about PopRep and we have used this software other times. Even for this dataset. However, results do not significantly differ which we think derives from the high levels of pedigree completness. Furthermore, the problem with PopRep is that data need to be hosted in an external repository which is often challenging when data protection policies exist.

21) If you have access to SNP or microsatellite data (from parentage testing) you could conduct STRUCTURE analysis to see if there really is population substructure linked to coat colour based on DNA evidence. Your paper relies on studbook / pedigree records that by your own admission are flawed and thus far from 100% reliable. Always a problem with studbook records because of human error be that intentional or not. PopRep does have good error checking routines that may pick up things that Endog does not.

Response: Although we admit this would be interesting, we do not have access to such information. Still, we think it is the combination of both which is really reliable and methods should be tailored in the specific context of each population. Furthermore, we do know of the challenges of carrying diversity studies when using pedigree as a source of information. However, SNPs or microsatellite have also problems which are linked to the lack of knowledge of the historic allele frequencies present in the population is hindered as molecular techniques may not distinguish the identity by descent (IBD) from identity by state (IBS) probabilities which underlie genetically mediated similarities among relatives. This produces a bias in diversity estimations due to genetic drift or unknown bottlenecks occurring throughout the history of a population.

22) Lacey et al point out the dangers of a breed management scheme that directly minimised either inbreeding or coancestry in the next generation. Better to manage inbreeding indirectly through managing mean kinship first and foremost for sustainable breeding. The aim is to increase effective population size by slowly reducing the rate of increase in inbreeding. Any sudden decrease in one generation can lead to a rebound in future generations as you increase the frequency of not only rare alleles but the more common ones as well and bring them together in a way that they can not be separated out in later generations. (Oliehoek et al explain this well http://www.breedingfordiversity.com)

Response: We agree, however, what we refer in the manuscript are the policies carried in this specific population as a result of a crossing between two ancestral breeds which do not equally contribute on every crossing (percentage of Arabian blood contribution may widely vary). Still, as addressed in the website proposed by the reviewer using mean kinship has also certain flaws which may make its used, at least discussable. For instance, after mean kinship is calculated for each individual, it is still not clear how breeding should follow. The mean kinship per animal does not indicate the number of offspring one animal should have, nor whether an animal should be selected. This is due to the combination of two properties (1) the mean kinship level does not show in what way an animal is related to the population. A higher mean kinship can be achieved by either a moderate relationship to the entire population or a very high relationship to a large part of the population. Thus, mean kinship does not always optimize genetic diversity, it depends on the population structure and (2) the current generation will differ from the next generation.

Round 2

Reviewer 2 Report

Your paper has been much improved by recent revisions.

Your figures are now clear and legible and with a colour legend that is consistent across figures.

Your use of plain English is much improved although I did detect some very minor typos and grammatical queries that you will probably want to sort out before publication:-

Line 125 typo should read "due to"

Line 149 would read better if edited as "A knowledge of the mechansims of inheritance of coat colour is useful for ........"

Line 165 patterns

Line 281 of genetic

Line 282 Nei's genetic distance was used ( if only one method) or Nei's genetic distances were used (if more than one method)